# Effects of Resuscitation and Simulation Team Training on the Outcome of Neonates with Hypoxic-Ischemic Encephalopathy in South Tyrol

**DOI:** 10.3390/jcm14030854

**Published:** 2025-01-28

**Authors:** Alex Staffler, Marion Bellutti, Arian Zaboli, Julia Bacher, Elisabetta Chiodin

**Affiliations:** 1Division of Neonatology/NICU, Hospital of Bolzano (SABES-ASDAA), Teaching Hospital of Paracelsus Medical University (PMU), 39100 Bolzano, Italy; marion.bellutti@sabes.it (M.B.); elisabetta.chiodin@sabes.it (E.C.); 2Innovation, Research and Teaching Service (SABES-ASDAA), Teaching Hospital of Paracelsus Medical University (PMU), 39100 Bolzano, Italy; arian.zaboli@sabes.it; 3Dornbirn City Hospital, Training Program for General Medicine, 6850 Dornbirn, Austria; julia.scm@hotmail.de

**Keywords:** simulation, team training, perinatal asphyxia, hypoxic ischemic encephalopathy

## Abstract

**Background/Objectives**: Neonatal hypoxic-ischemic encephalopathy (HIE) due to perinatal complications remains an important pathology with a significant burden for neonates, families, and the healthcare system. Resuscitation and simulation team training are key elements in increasing patient safety. In this retrospective cohort study, we evaluated whether regular constant training of all personnel working in delivery rooms in South Tyrol improved the outcome of neonates with HIE. **Methods**: We retrospectively analyzed three groups of neonates with moderate to severe HIE who required therapeutic hypothermia. The first group included infants born before the systematic introduction of training and was compared to the second group, which included infants born after three years of regular training. A third group, which included infants born after the SARS-CoV-2 pandemic, was compared with the previous two to evaluate retention of skills and the long-term effect of our training program. **Results**: Over the three study periods, mortality decreased from 41.2% to 0% and 14.3%, respectively. There was also a significant reduction of patients with subclincal seizures detected only through EEG, from 47.1% in the first period to 43.7% and 14.3% in the second and third study periods, respectively. Clinical manifestations of seizures decreased significantly from 47.1% to 37.5% and 10.7%, respectively, as well as severe brain lesions in ultrasound (US) and MRI. **Conclusions**: In this study, constant and regular simulation training for all birth attendants significantly decreases mortality and improves the outcome in neonates with moderate to severe HIE. This positive effect seems to last even after a one-year period during which training sessions could not be performed due to the COVID-19 pandemic.

## 1. Introduction

Hypoxic-ischemic encephalopathy (HIE) occurs in approximately 1.5 per 1000 live births in developed countries and is caused by a lack of oxygen and blood perfusion in the brain during the perinatal period [1,2]. The global burden is still higher, accounting for approximately one million deaths annually since the impact in low- and middle-income countries is increased compared to high-income countries [3,4,5].

The major causes are intrapartum hypoxic events and involve oxidative stress, excito-toxicity, failure of mitochondrial energy production, and cell death [6]. In addition, acute and chronic inflammation following hypoxemia-ischemia may lead to secondary cell loss [7]. Hypoxic insults are also associated with an increased brain temperature, likely due to increased metabolic demands after the acute injury [8]. Lowering core body temperature by 1 °C is reported to result in a 6–10% reduction in body metabolic demands [9].

Therapeutic hypothermia for 72 h to a core temperature of 33.5 °C is a standard therapy to reduce mortality and long-term neurological consequences [10,11,12]. This treatment reduces the rate of death or moderate to severe disability at 18–22 months from 62% in untreated to 41% in treated infants [13]. It has proven effective especially in moderate to severe HIE if started as early as possible within 6 h [14,15,16,17]. To guarantee its appropriate use, the American Academy of Pediatrics published recommendations and a framework of hypothermia [18]. Despite this neuroprotective treatment, mortality and lifelong morbidity, such as infantile cerebral palsy, cognitive delay, epilepsy, and learning and behavioral difficulties, remain high [1,19,20], with approximately 40–50% of newborns who have undergone hypothermic treatment presenting neurological difficulties in their further development [21,22,23].

Hypoxia and ischemia occur either before birth (20%), during birth (30%), before and during birth (35%), and, only in a few cases, after birth (10%) [24]. Such episodes are considered perinatal and neonatal emergencies, which always involve a multidisciplinary team. High-fidelity simulation training for obstetric emergencies can improve the skills of healthcare staff by regularly practicing emergency events without posing risk to patients [25]. It is proven that neonatal resuscitation and simulation training can significantly reduce the incidence of HIE and improve short term outcomes [26,27].

Since 2013, the Patient Safety Center in our institution has offered neonatal resuscitation and simulation team training for all staff working in delivery rooms in all hospitals in our region of South Tyrol. Almost all resuscitation and simulation instructors are doctors, nurses, or midwives working in our hospital, which is the reference perinatal center for the whole region. Regionalization of neonatal care took place in the 1990s. Since then, our NICU staff have been responsible for neonatal transportation and intensive care treatment of all neonates within the region. The purpose of this study is to examine the effects of this training on the outcomes of newborns with HIE.

## 2. Materials and Methods

This was a retrospective study of babies born during the period between October 2009 and December 2012, between January 2016 and December 2019, and between January 2021 and March 2024. All neonates with a diagnosis of HIE who were treated with therapeutic hypothermia during these periods were included. Five newborns were excluded due to underlying genetic pathologies or conditions. Included patients were divided into three cohorts (Figure 1):-Pre-training: October 2009–December 2012, N = 17-Post-training: January 2016–December 2019, N = 16-Post-SARS-CoV-2: January 2021–March 2024, N = 28

Patients in the three years between the first two cohorts (2013–2015) were excluded since this time was necessary to assure that almost all healthcare staff attended resuscitation and simulation training. These data were extrapolated by attendees’ lists, which were kept throughout the training periods. In the second period, we included patients until the breakout of the SARS-CoV-2 pandemic. We also excluded those patients born during the SARS-CoV-2 pandemic (December 2019–December 2020), since no courses could be organized during this period and because resuscitation wearing personal protective equipment could have a negative effect on resuscitation, as shown on manikin models [28]. The third cohort included patients with HIE born after the COVID-19 pandemic to evaluate skills retention and the related long-term effects of our training program. The timeline of each period and the number of included patients is shown in Figure 1.

### 2.1. Neonatal Resuscitation and Simulation Team Training

Between 2013 and 2019, and again from 2021 onwards, approximately 20 neonatal resuscitation and simulation team training sessions for neonatal and obstetric emergencies were organized annually at the Patient Safety Center in Bolzano. Participation in the courses was on a voluntary basis, but every member of staff working in this area was expected to take part at least once a year. The target groups were doctors in pediatrics, gynecology and obstetrics, neonatology, and anesthesia, as well as midwives and nurses.

All multidisciplinary training sessions were performed by 4 experienced instructors and included 12–15 participants. In these eight-hour courses, theoretical knowledge and technical and non-technical skills were all regularly addressed. These multidisciplinary trainings in obstetrics involved CTG interpretation, fetal distress detection, and scenarios involving critical situations that may occur during labor, including shoulder dystocia and emergency cesarean section. Neonatal resuscitation courses were based on the resuscitation guidelines of the AHA [29,30,31]. In addition to neonatal resuscitation courses, multidisciplinary training of high-fidelity scenarios focused on critical postnatal situations were regularly performed.

### 2.2. Patients and Outcomes

All neonates born in any hospital of our region and who met criteria for therapeutic hypothermia were admitted to the Neonatal Intensive Care Unit (NICU) in our Center and underwent hypothermic treatment within 6 h from the hypoxic ischemic event.

Criteria for the active cooling, according to the Guidelines of the Italian Society of Neonatology (SIN), were gestational age > 35 weeks, birth weight > 1800 gms, pH of ≤7.0 or base excess ≥ −12 mmol/L in a sample of umbilical cord blood or arterial analysis within the first hour of life, an Apgar score < 5 at ten minutes or need for positive pressure ventilation at 10 min. In these neonates, early neurological signs reflecting the severity of HIE were routinely evaluated based on Sarnat and Sarnat scores [13,32]. Those presenting a moderate to severe form underwent cooling and were therefore included in the study [13,14,15].

Included patients were coded with a consecutive number (pseudonymized), and demographic, clinical, laboratory, and diagnostic data were analyzed. All birth-related data, such as place of birth (inborn or outborn), birth weight, gender, gestational age, pH-value, and base excess (BE) collected within the first 60 min of life were documented and analyzed.

Main outcomes were mortality within 30 days of life, clinical and electrophysiological seizures recorded on video and EEG (performed for at least 30 min before hypothermic treatment, during cooling, and after the rewarming phase), as well as brain ultrasound and MRI findings. These were classified according to the Barkovich score: 0 = normal findings, 1 = abnormal signal in basal ganglia or thalamus, 2 = Isolated damage in the cortex, 3 = damage in the cortex and basal ganglia, or 4 = damage in the entire cortex and basal ganglia [19].

Additional data included the need for mechanical ventilation and inotropic agents, creatinine levels, and the occurrence of thrombocytopenia or bradycardia during hypothermic treatment.

### 2.3. Statistical Analysis

Categorical variables were expressed as numbers and percentages, while continuous variables were reported as means with standard deviation (SD) or medians with interquartile range (IQR), depending on their distribution. Univariate comparisons between the three groups were conducted using ANOVA or the Kruskal–Wallis test for continuous variables, depending on the normality of the data. For categorical variables, the Chi-square test was used. For comparison of outcomes between two groups, Student’s *t*-test, Kruskal–Wallis test, Mann–Whitney test, or Chi-square test were used, depending on the type of variable considered.

In addition, Kaplan–Meier survival curves were generated to compare 30-day mortality among the three groups, and comparisons between both two and three groups were made using the log-rank test. Given the small number of patients enrolled in the study, the statistical power of the log-rank test was calculated for pairwise group comparisons. The power was calculated assuming an alpha value (two-sided) of 0.05 and using various expected hazard ratios (HR) based on the observed mortality. All statistical analyses were performed using STATA software version 16.1. A *p*-value < 0.05 was considered statistically significant.

## 3. Results

A total of N = 61 patients were enrolled during the study period. Characteristics of the infants in the three cohorts are presented in Table 1. The groups differed only by BE value < 60 min, showing that severity of acidosis decreased progressively over the three phases of the study.

During the three study periods, mortality at 30 days decreased from 41.2% in the pre-training group to 0% in the post-training and 14.3% in the post-SARS-CoV-2 group (*p* = 0.007).

HIE scores were progressively less severe in the three study groups and normalities at head ultrasounds and at MRI of the brain increased significantly.

The incidence of subclinical seizures recorded only by EEG decreased, as well as clinical seizures.

Brain ultrasound (US) showed a significant increase in normal findings and a reduction in moderate and severe lesions. Echographic diagnoses were supported by MRI results, which showed a significant decrease of moderate and severe brain lesions according to Barkovich score [33]. Results are shown in Table 2.

Comparison between groups for mortality at thirty days was performed using Kaplan–Meier, as shown in Figure 2. Survival between the three groups shows a statistically significant difference with a Log-rank test reporting a *p* = 0.005. If we compare the pre-intervention group with the post-intervention group, the Log-rank test reports a *p* = 0.004, while if we compare the post-intervention group with the post-SARS-CoV-2 group, the Log-rank test reports a *p* = 0.119. The power of the log-rank test for pairwise comparisons was calculated by considering a variable HR for each comparison. For the comparison between the pre-training group and the post-training group (n = 33), with an expected HR of 0.35, a power of 0.8 was achieved. For the comparison between the post-intervention group and the post-COVID-19 group (n = 44), with an expected HR of 0.5, the power was 0.6. Finally, the comparison between the pre-intervention group and the post-COVID-19 group (n = 45), with an expected HR of 0.35, showed a power of 0.9.

Short term morbidity and initial interventions are presented in Table 3. The groups varied statistically significantly for the requirement of inotropic support, the incidence of mechanical ventilation, creatinine level, and incidence of thrombocytopenia. Incidence of bradycardia increased significantly across the three different study periods (<0.001).

## 4. Discussion

HIE after perinatal asphyxia is an important burden for infants and families due to the elevated associated mortality and morbidity [19]. Even though the mortality of fetuses, newborns, and mothers has been greatly reduced in recent decades, further reduction remains an important goal in perinatal care [34].

In this retrospective cohort study, we evaluated the possible effects of neonatal resuscitation courses combined with simulation team-training in obstetrics and neonatology for all doctors, midwives, and nurses working in the delivery rooms of all hospitals in our region.

Introduction and evaluation of such a training program involving all personnel was possible because the region is comparatively small, covering approximately 5000 neonates per year. In addition, all hospitals are part of the same organization, with our NICU being the perinatal center where all neonates at risk for HIE are cared for.

High rates of morbidity and mortality are potentially preventable and multidisciplinary team training could contribute to achieving this goal [35]. Resuscitation training and simulation team training are considered as potential quality improvement (QI) opportunities in neonates with HIE [36]. One previous study reported that simulation training of healthcare staff in neonatal resuscitation led to a reduction in neonatal mortality but also suggested the need of further studies to investigate a reduction in morbidity, including HIE and neurodevelopmental impairment [21]. Training in obstetric emergencies reportedly decreased HIE, but did not include medium-term clinical outcomes [22]. Our results show a significant reduction in mortality in the first month of life combined with a decrease in important morbidities like the incidence of moderate and severe HIE, a reduction of seizures, as well as more stable respiratory and cardiocirculatory conditions.

To our knowledge, this study is the first to report a significant reduction in mortality and a concomitant improvement in clinical outcome in a whole region. The improved neurologic outcome was in line with imaging on brain US and MRI. In addition, we found an improved respiratory and cardiovascular situation, with less need for invasive mechanical ventilation and less need for inotropic agents. The improved cardiovascular situation is also reflected by lower creatinine levels as a marker of better renal perfusion and better organ function. Regarding potential side effects of hypothermic treatment, we did not see a difference in thrombocytopenia but we did register an increase in bradycardia. This is explained because we introduced analgo-sedation with Dexmedetomidine, which has a proven effect on heart rate in these neonates [37].

Unfortunately, in this study, we cannot take into account specific details like personnel turnover and its possible influence on the experience of the single operators, although we are not aware of an unexpected high turnover in the departments of obstetrics, neonatology, and pediatrics during the three study periods. Other points that could not be addressed in this study were whether changes in neonatal resuscitation protocols, medications, or resuscitation conditions during the 15-year period might have influenced the outcomes, though we are not aware of any major changes that could have had such an impact. Since the region is small and all hospitals are part of the same organization, we can also confirm that resuscitation conditions in terms of spaces and materials did not change over the years in each hospital.

One weak point of this study is the relatively limited number of infants due to the small region and the low incidence of HIE (0.5–3/1000 live born neonates in HIC [2,19]). On the other hand, this is also a strength of this study, since we are well aware of all neonates requiring therapeutic hypothermia, which guarantees that we included all infants in the whole region. In addition, the personnel working in the hospitals of our region are all part of the same institution. Therefore, it is possible to define a program that guarantees neonatal resuscitation and simulation training for all health care providers working in the delivery room.

An important limitation of the study is the small sample size in each group, which affected the statistical power of the tests conducted in the study. Nevertheless, the power calculation for 30-day mortality yielded good results, with insufficient power observed only in the comparison between the post-training group and the post-SARS-CoV-2 group. However, this comparison is of limited importance, as the relevant comparison was against the baseline.

Future studies with larger samples will be necessary to confirm the observed results and improve the statistical power of our analyses. In addition, future research should focus on evaluating linear trends over time to better understand the progression of outcomes. Despite its limitations, the findings of this study provide important clinical insights into the effectiveness of the interventions adopted during the study periods.

## 5. Conclusions

Structured and regular neonatal resuscitation and simulation team-training for all birth attendants in the field of obstetrics and neonatology are associated with an improved outcome in neonates with HIE. This effect seems to persist even over a one-year period in which training could not be performed. Due to the small number of patients included, future studies with a larger number of patients are needed to confirm the results of the present study. Despite the abovementioned limitations, this study provides important additional information about the possible effects of resuscitation and team training on clinical outcomes.

## Figures and Tables

**Figure 1 jcm-14-00854-f001:**
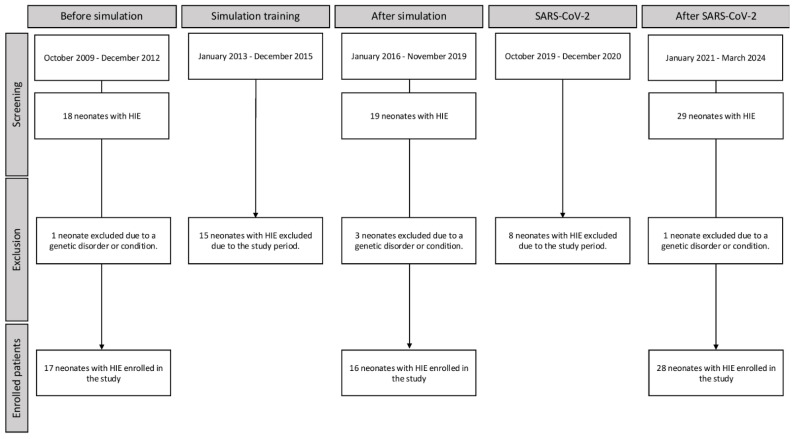
Timeline of interventions and evaluation periods.

**Figure 2 jcm-14-00854-f002:**
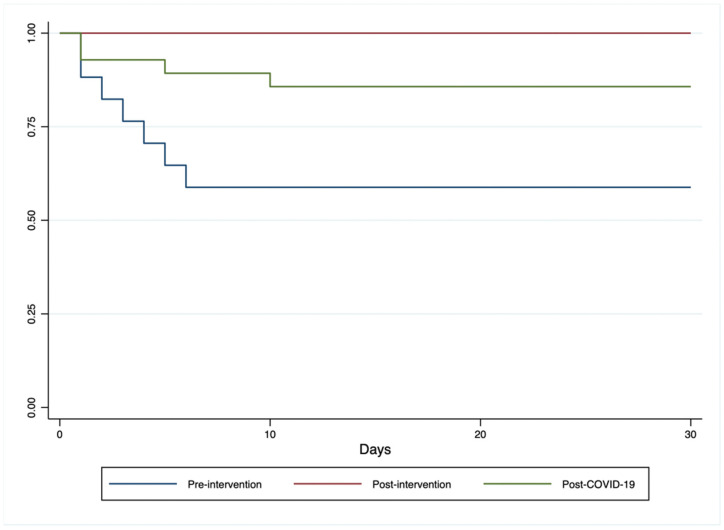
Kaplan–Meier curves showing mortality during the first 30 days of life in the three groups.

**Table 1 jcm-14-00854-t001:** Characteristics of included infants.

	Pre-Training	Post-Training	Post-SARS-CoV-2	*p*
Patients, n (%)	17 (27.9)	16 (26.2)	28 (45.9)	
Male, n (%) Female, n (%)	10 (58.8) 7 (41.2)	8 (50.0) 8 (50.0)	13 (46.4) 15 (53.6)	0.720
Inborn Outborn	5 (29.4) 12 (70.6)	8 (50.0) 8 (50.0)	8 (28.6) 20 (71.4)	0.311
Birth weight (gms), mean (SD)	3354 (512)	3119 (696)	3188 (505)	0.306
GA ^+^, mean (SD)	39.4 (1.6)	38.5 (2.4)	39.1 (1.7)	0.382
Term n (%) Preterm ^#^ n (%)	16 (94.1) 1 (5.9)	14 (87.5) 2 (12.5)	25 (89.3) 3 (10.7)	0.798
pH ^, mean (SD)	6.8 (0.2)	6.8 (0.2)	6.9 (0.2)	0.106
BE ^, mean (SD)	22.5 (7.2)	19.0 (6.2)	15.3 (5.9)	0.003 *

^+^ GA: gestational age; ^#^ defined as infants born between 35–37 weeks of gestational age; ^ pH and base excess reflect the lowest value during the first hour of life. * statistically significant.

**Table 2 jcm-14-00854-t002:** Mortality and neurological outcome.

	Pre-Training	Post-Training	Post-SARS CoV-2	*p*
Death < 30 days, n (%)	7 (41.2)	0 (0.0)	4 (14.3)	0.007 *
HIE score, n (%) Moderate Severe	6 (35.3) 11 (64.7)	8 (50.0) 8 (50.0)	21 (75.0) 7 (25.0)	0.093
EEG seizures, n (%)	8 (47.1)	7 (43.7)	4 (14.3)	0.032 *
Clinical seizures, n (%)	8 (47.1)	6 (37.5)	3 (10.7)	0.019 *
Brain US, n (%) Normal Hyperechogenic areas Status marmoratus	5 (29.4) 11 (64.7) 1 (5.9)	5 (31.2) 11 (68.8) 0 (0.0)	20 (71.4) 6 (21.4) 2 (7.1)	0.011 *
MRI, n (%) ^+^ 0 1 2 3	3 (30.0) 4 (40.0) 2 (20.0) 1 (10.0)	5 (31.2) 2 (12.5) 7 (43.7) 2 (12.5)	20 (80.0) 2 (8.0) 1 (4.0) 2 (8.0)	0.005 *

HIE: hypoxic ischemic encephalopathy; EEG: electroencephalogram; US: ultrasound, MRI: magnetic resonance tomography; ^+^ this result was obtained in n = 55/61 infants, and there were no infants with 4th grade lesions. * statistically significant.

**Table 3 jcm-14-00854-t003:** Short term morbidity and interventions.

	Pre-Training	Post-Training	Post-SARS-CoV-2	*p*
Need for mechanical ventilation, n (%)	17 (100.0)	11 (68.7)	14 (50.0)	0.002 *
Need for inotropic agents, n (%)	13 (76.5)	5 (31.2)	14 (50.0)	0.032 *
Creatinine > 1.2 mg/dL, n (%)	9 (56.2)	2 (12.5)	3 (11.5)	0.002 *
Bradycardia, n (%)	4 (23.5)	9 (56.2)	24 (85.7)	<0.001 *
Trombocytopenia, n (%)	8 (47.1)	5 (31.2)	3 (10.7)	0.023 *

* statistically significant.

## Data Availability

No data are shared due to privacy regulatory reasons.

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
