# Peer review of "Effects of Resuscitation and Simulation Team Training on the Outcome of Neonates with Hypoxic-Ischemic Encephalopathy in South Tyrol"

_jcm, 2025, doi:10.3390/jcm14030854_

Round 1
Reviewer 1 Report
Comments and Suggestions for Authors
The article Effects of resuscitation and simulation team training on the outcome of neonates with hypoxic-ischemic encephalopathy in South Tyrol discusses the effects of resuscitation and simulation team training. Recommendations:
1. The introduction should provide a more detailed discussion about neonates with hypoxic-ischemic encephalopathy.
2. Why were the respective time periods chosen? The 15-year gap between 2009 and 2024 implies significant differences in equipment, protocols, medications, resuscitation conditions, and personnel changes, which could influence the experience of the teams. Please explain. The explanation in lines 66–68 does not address these questions.
3. Why was a complete flowchart, including both included and excluded cases, not provided? Figure 1 is insufficient.
4. Differences between resuscitation teams could be notable depending on the center they work in and their level of experience. These details are not mentioned.
5. At line 124, what is the confidence interval? It should also be included in the tables.
6. In the results section, do not repeat the information already presented in the tables. Indicate the tests used below each table. Statistically significant results must be appropriately marked.
7. Lines 137–138: Percentages are not always relevant. % does not necessarily reflect statistical significance.
8. Where is Table 2 cited in the text?
9. In Table 2, statistical significance is observed, but this may be due to the small sample size, and post-training results include values of 0. Calculate the power of the study to demonstrate the relevance of the figures. This also applies to the Kaplan-Meier analysis. Why were multinomial regressions not included?
10. The discussion section needs to be expanded. First, discuss that neonatal resuscitation and mortality may also be influenced by other factors, such as neonates being transferred from less-equipped centers. Recommend: 10.3390/healthcare11243131. Additionally, recent studies show that pandemic periods and adrenaline doses significantly impact the resuscitation process. Recommend: https://doi.org/10.3390/jcm13237399.
11. The conclusions are too brief.
12. The references section requires more recent articles.
Author Response
- The introduction should provide a more detailed discussion about neonates with hypoxic-ischemic encephalopathy.
Authors: Thank you for the suggestion. In the introduction we added more detailed information about neonates with hypoxic-ischemic encephalopathy, especially the global burden, the pathophysiology and more information about treatment with therapeutic hypothermia.
- Why were the respective time periods chosen? The 15-year gap between 2009 and 2024 implies significant differences in equipment, protocols, medications, resuscitation conditions, and personnel changes, which could influence the experience of the teams. Please explain. The explanation in lines 66–68 does not address these questions.
Authors: Thank you for the suggestion. We specified better, how the time periods were chosen in the materials and methods section and discussed the other mentioned points in the discussion section.
- Why was a complete flowchart, including both included and excluded cases, not provided? Figure 1 is insufficient.
Authors: Thank you for this comment. We added a flowchart with included and excluded cases during the three study periods in the materials and methods section.
- Differences between resuscitation teams could be notable depending on the center they work in and their level of experience. These details are not mentioned.
Authors: Thank you for this suggestion. Training sessions involved all personnel working in the delivery rooms of all hospitals of the region, but the aim was not to compare resuscitation teams and their performance. Although we know for sure that the conditions under which the teams work and the materials used are the same throughout all hospitals, this study was not powered to detect any differences in the performance of involved teams or individual performance based on their level of experience. We addressed these points in the discussion section.
- At line 124, what is the confidence interval? It should also be included in the tables.
Authors: Thank you for your suggestion. Considering the total sample size (n = 61) and its division into three relatively small groups (17, 16, and 28 patients), we believe that including confidence intervals would not provide additional meaningful information due to the inherent variability associated with such small sample sizes.
Our decision to report variability using standard deviations (SD) for means and interquartile ranges (IQR) for medians reflects a scientifically rigorous and appropriate approach for descriptive statistics in this context. Confidence intervals, while valuable in larger datasets, would likely lead to overly wide ranges in our case, reducing their interpretative utility and potentially distracting from the main findings.
We believe this approach ensures clarity and accuracy in reporting, while remaining consistent with accepted statistical practices for small sample size.
- In the results section, do not repeat the information already presented in the tables. Indicate the tests used below each table. Statistically significant results must be appropriately marked.
Authors: Thank you for the comment. We removed redundant information in this section. In addition, we marked statistically significant values in tthe tables accordingly.
- Lines 137–138: Percentages are not always relevant. % does not necessarily reflect statistical significance.
Authors: Thank you for this comment. A p-value was inserted, in order to make statistical significance clearer to the reader.
- Where is Table 2 cited in the text?
Authors: Thank you for the suggestion, we had missed that previously. We inserted the citation of Table 2 in the text.
- In Table 2, statistical significance is observed, but this may be due to the small sample size, and post-training results include values of 0. Calculate the power of the study to demonstrate the relevance of the figures. This also applies to the Kaplan-Meier analysis. Why were multinomial regressions not included?
Authors:Thank you for the comments and observations. Below, we address the points raised:
- We acknowledge that the sample size is limited, which may affect the ability of the study to detect significant differences. We calculated the statistical power for Kaplan-Meier analyses and the log-rank test. The results show sufficient power (≥80%) for some comparisons, such as between the Pre-training and Post-SARS-CoV-2 groups, but it remains insufficient for others, such as between the Post-training and Post-SARS-CoV-2 groups. However, the latter comparison has limited impact on the study, as the key result lies in the comparisons with the baseline (Pre-training group).
An explicit power calculation was not performed for the univariate analysis due to the limitations in the number of observed events (only 11 deaths out of 61 patients).
These considerations have been included in the manuscript in the methods, results, and study limitations sections.
- Multinomial regression was not included as it is not applicable to the study design. The primary outcome of the study is dichotomous (30-day mortality: yes/no), while the three study groups (Pre-intervention, Post-intervention, Post-COVID-19) are explanatory independent variables. Furthermore, the goal was not to create a predictive model but to assess the impact of training on mortality. In this context, an appropriate method for analyzing a dichotomous outcome, Kaplan-Meier curves and the log-rank test, was used.
Multinomial regression would have been appropriate only if the outcome had more than two nominal categories (e.g., "recovery," "deterioration," "death"), which is not the case in this study.
- We understand the concern about the impact of the small sample size on the statistical significance observed in Table 2. To mitigate this issue: Kaplan-Meier analyses and the log-rank test were used to verify whether there were significant differences in 30-day mortality between the groups. In the Post-intervention group, the observed value of 0 deaths highlights a potential benefit associated with the intervention. However, this observation should be interpreted with caution due to the small sample size and limited number of events.
- The discussion section needs to be expanded. First, discuss that neonatal resuscitation and mortality may also be influenced by other factors, such as neonates being transferred from less-equipped centers. Recommend: 10.3390/healthcare11243131. Additionally, recent studies show that pandemic periods and adrenaline doses significantly impact the resuscitation process. Recommend: https://doi.org/10.3390/jcm13237399.
Authors: Thank you for these suggestions. Indeed, the referral of neonate from less equipped to higher equipped units may influence mortality, though during all three study periods the referrals did not change in terms of indications for transfer, stabilization before transport or transport equipment (neither personnel, nor equipment). Therefore, we would not expect any impact of transportation in our study cohorts. We also included a short paragraph on Regionalization of neonatal care and the responsibility of our NICU staff in transport and intensive care of neonates in our region. We added the possible difficulty of resuscitation wearing PPEs during pandemic periods.
- The conclusions are too brief.
Authors: Thank you for this comment. We expanded the conclusions section underlining the low number of included patients as a weakness of the study.
- The references section requires more recent articles.
Authors: Thank you for this suggestion, we updated the references section including more recent articles.
Reviewer 2 Report
Comments and Suggestions for Authors
I have read this paper with a background on perinatal clinical research, including the topic of therapeutic hypothermia and perinatal asphyxia. While I do value the paper as reported, I do have some concerns and uncertainties to share
Firstly, the language is too causal. What the authors describe are association, not necessary causal, as perhaps not the training in itself, but ‘Hawthorne’ type of effects are involved. In this way, effects of training are eg too causal in my assessment.
Secondly, have the authors considered more continuous analysis like ANOVA type of analysis, with the initiation of the training, or finalization of training as pivotal effect (trends over years affected or not by the training intervention).
Finally, can you provide some more information on the training. Was this restricted to postnatal scenario’s, or also emergency caesarean and/or fetal distress detection?
Specific
Line 17, increased the outcome, do you mean improve ?
Line 38-39, to the best of my understanding, TH is applied to moderate to severe cases, suggest to add this. This is relevant, as I was somewhat surprised that the guidelines to initiate TH do not mention neurological impairment, while you do mention somewhat lower the Sarnat score. Could you please check these sentences and provided additional detailed information on this (Lines 89-96, versus Line 100-101).
Table 3, prolonged coagulation time ?
I found the final lines of the discussion section revealing as quite interesting, as it seems it does provide the ‘volume’ needed to sustain the team of trainers. I would request some more information on the resources needed to do so ?
Author Response
Firstly, the language is too causal. What the authors describe are association, not necessary causal, as perhaps not the training in itself, but ‘Hawthorne’ type of effects are involved. In this way, effects of training are eg too causal in my assessment.
Authors: Thank you for this comment. We revised the manuscript and changed the language, in order to make the description of the results less causal. In addition, the conclusion section was changed accordingly, in order to clarify the low number of included patients as a weakness of our study. All changes were made visible in the text.
Secondly, have the authors considered more continuous analysis like ANOVA type of analysis, with the initiation of the training, or finalization of training as pivotal effect (trends over years affected or not by the training intervention).
Authors: Thank you for the suggestion. We considered the idea of a continuous analysis to evaluate temporal trends related to the training interventions. However, our dataset does not allow for this type of analysis for the following reasons: 1. There were time periods without any patients included in the study (e.g., during the training period and the SARS-CoV-2 phase), which prevent a continuous temporal analysis or models that assess linear trends. 2. The study groups were defined as distinct periods (before training, after training, and post-SARS-CoV-2), with limited numbers of patients in each group. This design is more suited to categorical comparisons between groups rather than longitudinal analysis.
A continuous analysis would require uninterrupted data collection and a larger sample size for each period, but our study does not meet these conditions. Despite these limitations, the methods used are appropriate for the study design and address the research question. That said, we have included in the limitations section the need for future studies that could explore temporal trends with larger and more complete datasets.
Finally, can you provide some more information on the training. Was this restricted to postnatal scenario’s, or also emergency caesarean and/or fetal distress detection?
Authors: Thank you for this suggestion. We described the training contents in more detailed way, including examples of obstetric and neonatal scenarios. We also specified that emergency caesarean section and fetal distress detection, as well as CTG evaluation and interpretation were specifically addressed in the trainings.
Specific
Line 17, increased the outcome, do you mean improve?
Authors: Thank you very much, we changed the term “increased” to “improved”.
Line 38-39, to the best of my understanding, TH is applied to moderate to severe cases, suggest to add this. This is relevant, as I was somewhat surprised that the guidelines to initiate TH do not mention neurological impairment, while you do mention somewhat lower the Sarnat score. Could you please check these sentences and provided additional detailed information on this (Lines 89-96, versus Line 100-101).
Authors: Thank you for this comment. We specified that TH is applied to moderate and severe cases. We also clarified the sentences regarding the Sarnat and Sarnat score for the evaluation of the neurological condition of the neonates before undergoing TH.
Table 3, prolonged coagulation time?
Authors: Thank you for this question. We removed the value, since it does not add any important useful information to the manuscript.
I found the final lines of the discussion section revealing as quite interesting, as it seems it does provide the ‘volume’ needed to sustain the team of trainers. I would request some more information on the resources needed to do so?
Authors: Thank you for this comment. We included more detailed information about the resources required for the training sessions in the materials and methods section.
